# Association of Clinical Manifestations of Systemic Lupus Erythematosus and Complementary Therapy Use in Taiwanese Female Patients: A Cross-Sectional Study

**DOI:** 10.3390/medicina58070944

**Published:** 2022-07-17

**Authors:** Ming-Chi Lu, Chia-Wen Hsu, Hui-Chin Lo, Hsiu-Hua Chang, Malcolm Koo

**Affiliations:** 1Division of Allergy, Immunology and Rheumatology, Dalin Tzu Chi Hospital, Buddhist Tzu Chi Medical Foundation, Dalin, Chiayi 622401, Taiwan; e360187@yahoo.com.tw; 2School of Medicine, Tzu Chi University, Hualien City 97004, Taiwan; 3Department of Medical Research, Dalin Tzu Chi Hospital, Buddhist Tzu Chi Medical Foundation, Dalin, Chiayi 622401, Taiwan; chiawen0114@yahoo.com.tw (C.-W.H.); df289469@tzuchi.com.tw (H.-C.L.); df274760@tzuchi.com.tw (H.-H.C.); 4Graduate Institute of Long-Term Care, Tzu Chi University of Science and Technology, Hualien City 970302, Taiwan; 5Dalla Lana School of Public Health, University of Toronto, Toronto, ON M5T 3M7, Canada

**Keywords:** systemic lupus erythematosus, complementary medicine, clinical manifestations, probiotics, Raynaud’s phenomenon

## Abstract

*Background and Objectives*: Systemic lupus erythematosus (SLE) is a chronic systemic autoimmune disease that affects predominantly women in the childbearing years. Patients may seek complementary therapies to manage their health and to reduce symptoms. However, to our knowledge, no studies have explored the association between clinical manifestations of SLE and complementary therapies. Therefore, this study aimed to investigate the association of complementary therapies with common clinical manifestations in Taiwanese female patients with SLE. *Materials and Methods*: A cross-sectional study was conducted at a regional teaching hospital in southern Taiwan. Outpatients from the rheumatology clinic who met the inclusion criteria were consecutively recruited. Demographic data, clinical manifestations of SLE, and types of complementary therapy use were determined using paper-based questionnaire. Multiple logistic regression analyses were conducted to investigate the use of complementary therapies associated with clinical manifestations of SLE. *Results*: Of the 317 female patients with SLE, 60.9% were 40 years or older. The five SLE clinical manifestations with the highest prevalence were Raynaud’s phenomenon (61.2%), photosensitivity (50.2%), Sjögren’s syndrome (28.4%), arthralgia and arthritis (22.1%), and renal involvement (14.5%). Multiple logistic regression analyses revealed that Raynaud’s phenomenon was significantly associated with fitness walking or strolling (adjusted odds ratio [aOR] 1.77; *p* = 0.027) and fish oil supplements (aOR 3.55, *p* < 0.001). Photosensitivity was significantly and inversely associated with the use of probiotics (aOR 0.49; *p* = 0.019). Renal involvement was significantly associated with the use of probiotics (aOR 2.43; *p* = 0.026) and visit to the Chinese medicine department in a hospital (aOR 3.14, *p* = 0.026). *Conclusions*: We found that different clinical manifestations of SLE were associated with the use of different complementary therapies. Health care providers should have up-to-date knowledge of common complementary therapies and be ready to provide evidence-based advice to patients with SLE.

## 1. Introduction

Systemic lupus erythematosus (SLE) is a complex, chronic, systemic autoimmune disease that mainly affects women of childbearing age. The worldwide prevalence of SLE varied considerably from 2 to 7 per 10,000 people [1]. In Taiwan, the reported incidence of SLE in 2011 was 8.1 per 10,000 people with 14.3 and 1.6 per 10,000 women and men, respectively [2]. Many organs or systems can be involved, leading to multiple clinical manifestations in patients with SLE. In addition, poor body image, severe fatigue, and psychological morbidity could negatively affect health-related quality of life, resulting in a high prevalence of disability among patients with SLE [3,4].

Despite the substantial progress in the medical treatment of SLE, patients continue to live with a range of clinical symptoms, such as fatigue, joint pain, and skin rash [5]. In addition to conventional pharmacological therapies, patients may seek various complementary therapies to manage their chronic symptoms. While there is not yet a universal accepted operational definition of complementary therapies [6], it was estimated that more than half of patients with SLE had used complementary therapies to manage their health and to reduce symptoms [7]. A nationwide survey in the United Kingdom of 2527 people with SLE revealed that 32% of them sought complementary therapies, with acupuncture, massage, and vitamin supplements being the most commonly used [8]. Moreover, our previous study showed that over 85% of Taiwanese patients with SLE used complementary therapies on a regular basis. The top five popular types of complementary therapies used were fitness walking or strolling, Buddhist prayer or attending temples, vitamins, calcium supplements, and fish oil [9]. Nevertheless, while there is some evidence that certain complementary therapies could be beneficial for certain clinical conditions, studies on their efficacy and long-term safety are still limited [10].

Motivation to use complementary therapies in patients with SLE is complex. A study conducted in a tertiary-care rheumatology center in Singapore revealed that there were two types of users—those who intended to use complementary therapies to treat SLE and those who used them for general health maintenance [11]. Furthermore, a systematic review of randomized controlled trials of non-pharmacologic therapies, predominantly psychological interventions, in patients with SLE revealed that these therapies were significantly associated with an improvement in fatigue, anxiety and depression, and pain in some studies [12]. However, to our knowledge, there were no studies on the association between common clinical manifestations of SLE and the use of various complementary therapies. Therefore, the aim to this study was to investigate the association of prevalent SLE clinical manifestations and complementary therapy use in Taiwanese female patients with SLE.

## 2. Materials and Methods

### 2.1. Study Design and Population

This cross-sectional study was a sub-study of our previous investigation of factors associated with the use of complementary therapies among Taiwanese patients with SLE [9]. In the present study, we focused on the use of complementary therapy in female patients with different clinical manifestations of SLE. Only female patients were analyzed because the clinical manifestations of SLE are different between the sexes [13].

Outpatients attending the rheumatology clinic in a regional teaching hospital in southern Taiwan were consecutively recruited into the study between April and August 2019. A paper-based questionnaire was used to obtain information from the patients. Two rheumatology clinic research nurses were available to assist with the completion of the questionnaire, if necessary.

The sample size was estimated using G*Power software (version 3.1.9.4) [14]. For a multiple regression analysis with 20 predictors, α of 0.05, power of 90%, and Cohen’s *f*^2^ effect size of 0.09, 307 participants would be needed. The *f*^2^ was set at halfway between a small effect size (0.02) and a medium (0.15) effect size [15].

The study was conducted according to the guidelines of the Declaration of Helsinki, and all patients signed an informed consent. The study protocol was approved by the institutional review board of Dalin Tzu Chi Hospital, Buddhist Tzu Chi Medical Foundation (No. B10801017). 

### 2.2. Inclusion and Exclusion Criteria

The inclusion criteria for this study were female patients aged ≥ 20 years, those who met clinician-confirmed diagnosis of SLE according to the 1997 American College of Rheumatology revised criteria [16] or the 2012 Systemic Lupus International Collaborating Clinics Classification Criteria [17]. Patients who had previously been diagnosed with other important systemic autoimmune diseases, including rheumatoid arthritis, systemic sclerosis, spondyloarthritis, dermatomyositis, polymyositis, or juvenile idiopathic arthritis were excluded from the study.

### 2.3. Clinical Manifestations of Systemic Lupus Erythematosus 

The clinical manifestations of SLE were defined in advance and evaluated at the time of enrollment by attending physicians and research nurses. Common clinical manifestations were selected based on the definition of the SLE Disease Activity Index 2000 (SLEDAI-2K) [18] and their frequency of occurrence [19,20]. In addition, we chose those symptoms that are more likely to have a direct impact on patients’ quality of life. A total of 10 clinical manifestations were investigated in this study, including Raynaud’s phenomenon, photosensitivity, Sjögren’s syndrome, arthralgia or arthritis, renal involvement, malar rash, oral ulcer, alopecia, skin vasculitis, and discoid lesion. Manifestations such as leukopenia, thrombocytopenia, low complement, and elevated dsDNA level were omitted from this study because patients with SLE are less likely to seek complementary therapies that do not directly affect their quality of life. Clinical manifestations were considered present only when patients were experiencing them at the time of the survey.

### 2.4. Measurement of Demographic Variables 

The following demographic information was determined from the questionnaire: age, body mass index, educational level, marital status, employment status, self-perceived health status, age at the diagnosis of SLE, smoking habit, alcohol use in the past year, regular vigorous exercise in the past year, and daily duration of sleep. In this study, overweight (24 ≤ body mass index < 27 kg/m^2^) and obesity (body mass index ≥ 27 kg/m^2^) were defined according to the Ministry of Health and Welfare, Taiwan [21]. Regarding the question about exercise, the respondents were asked whether they had engaged in exercise that lasted at least 20 minutes that made them breathe faster and sweat in the past year. The response categories were never, once a month or fewer, several times a month, several times a week, and daily. The last two categories were combined and defined as regular vigorous exercise.

### 2.5. Use of Complementary Therapy 

Complementary therapies were presented as seven broad categories in the questionnaire, as described in our previous study [9] (Table 1). A Likert-type scale with four response choices was used. These four response categories were collapsed into two responses by treating the “always use” category as “use” while the remaining three categories (sometimes use, had tried previously, and never use) as “not use”.

### 2.6. Data Analysis

The basic characteristics of the study participants were summarized as frequencies with percentages. The 10 most popular types of complementary therapies were identified based on their frequency of use by patients in this study. The five SLE clinical manifestations with the highest prevalence were treated as outcome variables and analyzed using multiple logistic regression analysis. The independent variables were the 10 types of complementary therapies with the highest prevalence and all demographic variables listed in Table 1. A backward variable selection method based on likelihood ratios was used to obtain the final regression model. All statistical analyses were performed using IBM SPSS Statistics for Windows, version 27.0.1.0 (IBM Corp., Armonk, NY, USA). *p* < 0.05 was considered statistically significant.

## 3. Results

The basic characteristics of the 317 patients with SLE are summarized in Table 2. Most of them (60.9%) were 40 years or older. About 74% of the patients reported that their health status was average, poor, or very poor. The prevalence of the 10 clinical manifestations of SLE in our patients is shown in Figure 1. The five clinical manifestations with the highest prevalence were Raynaud’s phenomenon (61.2%), photosensitivity (50.2%), Sjögren’s syndrome (28.4%), arthralgia and arthritis (22.1%), and renal involvement (14.5%). Furthermore, the 10 most popular types of complementary therapies used by patients in this study were the following: fitness walking or strolling (37.5%), Buddhist chanting (37.2%), vitamins (31.9%), calcium supplements (24.6%), fish oil supplements (19.6%), probiotics (17.7%), massage therapy or Tui Na (11.7%), fitness workout (11.0%), visit to Chinese medicine clinics (8.8%), and visit to the Chinese medicine department in a hospital (7.9%).

The results of multiple logistic regression analyses of the clinical manifestations of SLE are shown in Table 2. Of the five SLE clinical manifestations with the highest prevalence, two of them (Sjögren’s syndrome, arthralgia or arthritis) were not significantly associated with the use of any of the 10 complementary therapies. Therefore, only the results of the remaining three clinical manifestations of SLE are shown in Table 3. First, Raynaud’s phenomenon was significantly associated with fitness walking or strolling (adjusted odds ratio [aOR] 1.77; *p* = 0.027) and fish oil supplements (aOR 3.55; *p* < 0.001), adjusted for age at diagnosis of SLE. Second, photosensitivity was significantly associated with probiotics (aOR 0.49; *p* = 0.019). Third, renal involvement was significantly associated with probiotics (aOR 2.43; *p* = 0.026) and visit to the Chinese medicine department in a hospital (aOR 3.14; *p* = 0.026), adjusted for body mass index, educational level, and marital status. 

## 4. Discussion

In this cross-sectional study of Taiwanese women with SLE, we reported the association between the use of complementary therapies and common clinical manifestations of SLE. Patients with SLE are often present with various systemic manifestations, and many of them are not SLE-specific, such as fatigue and fever. In our study, two of the five clinical manifestations with the highest prevalence, namely Sjögren’s syndrome and arthralgia or arthritis, were not associated with the use of any of the complementary therapies. 

In our female patients, the most common clinical manifestation of SLE with significant use of complementary therapies was Raynaud’s phenomenon, which affected 61.2% of them. The Raynaud phenomenon is a nonspecific skin manifestations of SLE, resulting from a vasospasm typically triggered by cold conditions or emotional stress. The present study showed that the presence of Raynaud’s phenomenon was associated with fitness walking or strolling and fish oil supplements. Although there is no specific research that evaluates the efficacy of exercise in the treatment of Raynaud’s phenomenon, a meta-analysis of 11 trials with a total of 355 participants revealed that exercise could significantly improve microvascular and macrovascular function in patients with autoimmune diseases [22]. In the present study, we observed a significant increased use of fitness walking or strolling in our patients, as low-impact may improve blood circulation and may thereby improve Raynaud’s phenomenon. We also noted that the use of fish oil supplements was significantly associated with the presence of Raynaud’s phenomenon. A prospective double-blind, randomized, control study using 32 patients with Raynaud’s phenomenon showed that taking fish oil could improve tolerance to cold exposure and delay the onset of vasospasm in patients with primary, but not secondary, Raynaud’s phenomenon [23]. However, the beneficial effect of fish oil in patients with secondary Raynaud’s phenomenon has not been validated in further studies. In addition, a number of food items, such as garlic, ginkgo biloba, L-carnitine, inositol nicotinate, and evening primrose oil had been reported to increase skin blood flow or hand skin temperature. Overall, evidence from rigorous studies is still lacking to support any food ingredients in alleviating Raynaud’s phenomenon [24]. Nevertheless, given the general beneficial health effects of fish oil on cardiovascular disease [25], and possibly on several autoimmune diseases, such as multiple sclerosis [26], rheumatoid arthritis [27], and psoriasis [28], future intervention studies on the association between fish oil and SLE should also include the evaluation of Raynaud’s phenomenon as an outcome [29]. 

The second most prevalent clinical manifestation of SLE with significant use of complementary therapies was photosensitivity. Photosensitivity is a highly complex condition and is a common clinical manifestation of SLE. Exposure to ultraviolet radiation can lead to increased skin disease flares and systemic symptoms, such as joint pain and fatigue [30]. In this study, the use of probiotics was significantly and inversely associated with the presence of photosensitivity. Previous research suggested that probiotics could potentially be used in the prevention and management of allergic diseases [31], allergic inflammation, skin hypersensitivity, and UV-induced skin damage [32]. As photosensitivity is often referred to sun allergy by the general population, it is surprised to observe an increased use of complementary therapies that are thought to be able to alleviate allergy reaction. However, the opposite association was observed in our study—patients with photosensitivity were associated with decreased use of probiotics. The reduction could possibly be related to the concern of stimulating the immune response by probiotics [33,34]. Animal studies have shown that probiotics could modify various immune parameters, such as the innate immune response of macrophages and dendritic cells [35], and the cell wall structure of probiotic *Lactobacillus casei* could potently induce IL-12 production [36]. In contrast, animal studies suggested that intake of *Lactobacillus casei* Shirota could alleviate SLE symptoms and their cardiovascular and renal complications [37]. More research is required to establish the safety and efficacy of probiotics for the prevention and treatment of photosensitivity in patients with SLE.

The third most prevalent clinical manifestation of SLE with significant use of complementary therapies was renal involvement. The kidney is the most commonly affected visceral organ in SLE, and renal failure and sepsis are two of the main causes of mortality in patients with SLE [38]. In the present study, the use of probiotics and visits to the Chinese medicine department in a hospital were significantly associated with renal involvement. A meta-analysis of 13 randomized controlled trials showed that the intake of probiotics, prebiotics, and synbiotics could reduce the formation of uremic toxin, p-cresol, and their serum levels [39]. Another meta-analysis of 13 clinical trials revealed that prebiotic, probiotic, and synbiotic supplementation could significantly decrease urea and blood urea nitrogen, but uric acid was increased. No significant changes in the glomerular filtration rate were observed [40]. However, no studies have specifically examined the effect of probiotics on renal function in patients with SLE. Furthermore, as safety reporting in studies assessing probiotics, prebiotics, and synbiotics is still inadequate [41], the potential risk in using probiotics in immunocompromised patients must be carefully evaluated.

In the present study, renal involvement was significantly associated with a more frequent visit to the Chinese medicine department in a hospital. A meta-analysis of six randomized controlled trials with 470 patients showed that the combination application of traditional Chinese and Western Medicine could improve the clinical efficacy of treatment of lupus nephritis with lower 24-hour urine protein, serum creatinine, and decrease adverse drug reactions [42]. Based on the secondary analysis of 16,645 newly diagnosed SLE patients identified from the Taiwan National Health Insurance Research Database, the combined use of conventional medicine and traditional Chinese medicine was found to significantly decrease the risk of lupus nephritis among Taiwanese patients with SLE [43]. Additional studies are warranted to explore the type of Chinese medicine prescriptions that were commonly used in SLE patients with renal involvement and their efficacy when combined with Western medicine. 

There were some limitations in this study. First, due to the cross-sectional design of the present study, causal inferences between the SLE manifestations and their associated factors could not be established. Second, the findings of the study might not be generalizable to other countries with different health care systems and cultural dimensions. 

## 5. Conclusions

In conclusion, the prevalence of SLE clinical manifestations and the use of complementary therapies were identified in female Taiwanese patients with SLE. It is important that health care providers have up-to-date knowledge of common complementary therapies and be ready to provide evidence-based advice to patients with SLE. Furthermore, given the increasing use of fish oil supplements for Raynaud’s phenomenon and probiotics for renal involvement, their safety and efficacy should be investigated in future studies. 

## Figures and Tables

**Figure 1 medicina-58-00944-f001:**
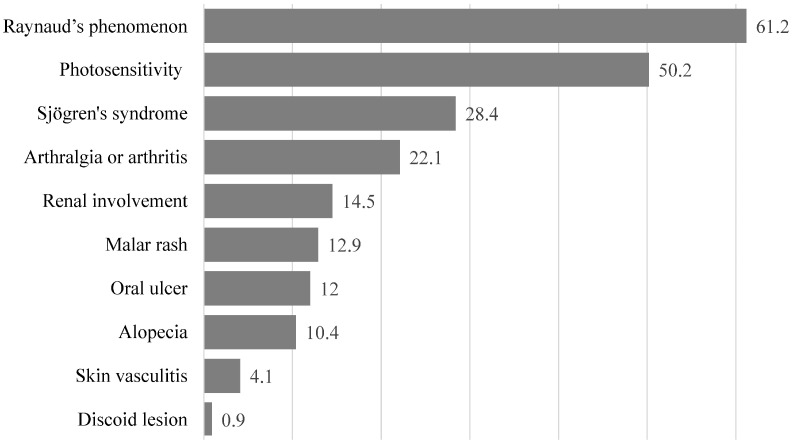
Prevalence of different clinical symptoms in Taiwanese female patients with systemic lupus erythematosus. Numbers shown are percentages.

**Table 1 medicina-58-00944-t001:** Categories of complementary therapies.

Complementary Therapy Category	Description
1. Body-based and energy therapy	massage therapy or Tui Na (Chinese massage), chiropractic or osteopathic manipulation, Gua Sha therapy or cupping, acupuncture or moxibustion, and far-infrared therapy
2. Mind-body therapy	qigong or Tai Chi, meditation, relaxation therapy, and aromatherapy
3. Folk remedies and religious practices	divination and nameology, exorcise, Buddhist chanting, and praying
4. Exercise therapy	dancing, fitness workout, jogging, fitness walking or strolling, swimming, and cycling
5. Chinese medicine	visit to the Chinese medicine department in a hospital, visit to Chinese medicine clinics, Chinese medicinal herbs shop, and herbal remedies
6. Nutrient supplements	vitamins, fish oil supplements, ginkgo, calcium supplements, glucosamine, turmeric, and probiotics
7. Diet therapy	raw food diet, organic diet, Mediterranean diet, low-carbohydrate diet, and ketogenic diet

**Table 2 medicina-58-00944-t002:** Basic characteristics of female patients with systemic lupus erythematosus (*n* = 317).

Variable	*n* (%)
Age interval, years	
20–39	124 (39.1)
≥40	193 (60.9)
Body mass index	
Normal	167 (52.7)
Underweight	45 (14.2)
Overweight or obese	105 (33.1)
Educational level	
High school or below	158 (49.8)
College or above	159 (50.2)
Marital status	
Single	106 (33.4)
Being married, widowed, or divorced	211 (66.6)
Employment status	
Employed	200 (63.1)
Unemployed	117 (36.9)
Self-perceived health status	
Very good and good	83 (26.2)
Average, poor and very poor	234 (73.8)
Age at SLE diagnosis, years	
20–29	172 (54.3)
≥30	145 (45.7)
Smoking habit	
No	296 (93.4)
Daily or occasionally	21 (6.6)
Alcohol use in the past year	
No	248 (78.2)
Daily or occasionally	69 (21.8)
Regular vigorous exercise in the past year	
No	169 (53.3)
Yes	148 (46.7)
Duration of sleep/day, hours	
≥8	59 (18.6)
≤7	258 (81.4)

**Table 3 medicina-58-00944-t003:** Multiple logistic regression analyses of factors associated with clinical manifestations with significant use of complementary therapies in Taiwanese female patients with systemic lupus erythematosus (*n* = 319).

Variable	Raynaud’s Phenomenon		Photosensitivity		Renal Involvement	
	Adjusted odds ratio (95% CI)	*p*	Adjusted odds ratio (95% CI)	*p*	Adjusted odds ratio (95% CI)	*p*
Body mass index						
Normal					1	
Underweight					2.67 (1.13–6.28)	0.025
Overweight or obese					1.61 (0.73–3.54)	0.237
Educational level						
High school or below					1	
College or above					2.64 (1.00–6.92)	0.049
Marital status						
Being married, widowed, or divorced					1	
Single					3.32 (1.70–6.46)	<0.001
Age at SLE diagnosis, years						
≥30	1					
20–29	1.76 (1.09–2.84)	0.022				
Fitness walking or strolling	1.77 (1.07–2.92)	0.027				
Fish oil supplements	3.55 (1.75–7.19)	<0.001				
Probiotics			0.49 (0.27–0.89)	0.019	2.43 (1.11–5.30)	0.026
Visit Chinese medicine department in a hospital					3.14 (1.15–8.58)	0.026

CI: confidence interval. In the multiple regression logistic model, the following variables were evaluated: age interval, body mass index, educational level, marital status, employment status, self-perceived health status, age at diagnosis of SLE, smoking habit, alcohol use in the past year, regular exercise in the past year, duration of sleep per day, and 10 types of complementary therapies (fitness walking or strolling, Buddhist chanting, vitamins, calcium supplements, fish oil supplements, probiotics, massage therapy or Tui Na, fitness workout, visit to Chinese medicine clinics, and visit to the Chinese medicine department in a hospital).

## Data Availability

The data that support the findings of this study are available on request from the corresponding author.

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
