# Peer review of "Association of Clinical Manifestations of Systemic Lupus Erythematosus and Complementary Therapy Use in Taiwanese Female Patients: A Cross-Sectional Study"

_medicina, 2022, doi:10.3390/medicina58070944_

Round 1
Reviewer 1 Report
Reviewer comments
Title
Association of Clinical Manifestations of Systemic Lupus 2 Erythematosus and Complementary Therapy Use in Taiwanese 3 Female Patients: A Cross-sectional Study
Overall, this is a clear, concise, and well-written manuscript. The introduction is relevant, and theory based. Overall, this is a high-quality manuscript that has implications; however, specific modifications are needed as the following:
|
Abstract: |
Well structured and written The conclusion is supported by the results |
||||||
|
Introduction |
Interesting for reading (1) fit-61 ness walking or strolling, (2) Buddhist prayer or attending temples, (3) vi…etc... It is better to delete numbering here as they are confusing with the reference number |
||||||
|
Materials and Methods Study design and population |
In the pre-80 sent study, we focused on the use of complementary therapies in female patients with 81 different SLE clinical manifestations…this is a repition for what have mentioned in the introduction |
||||||
|
Inclusion and exclusion criteria |
The inclusion of females only was not mentioned in the authors in the inclusion criteria Line 92.. aged 20..why the author choose this age and not 18 . 96.. major..I think this not a scientific term .. |
||||||
|
Clinical manifestations of SLE |
It will be more valuable to include the symptoms for which complementary treatment was needed The authors need to explain why they choose these 11 symptoms only Including dialysis as a common clinical manifestation is not appropriate |
||||||
|
Line 111 |
What is meant by age interval |
||||||
|
Line 113 |
alcohol use in the past year. The authors specify the last year. Why not the alcohol in general ..and what is meant by use ..chronic drinking or only one time or what ..this point is not clear |
||||||
|
|
The method of questionnaire distribution among patients was not mentioned and the method of fulfilling the questionnaire is not mentioned either by the patient or by the researcher Please determine if this questionnaire was validated or not Determine the language of questionnaire and if it was translated or not |
||||||
|
Line 145 |
average, poor, or very poor…. the authors considered average and poor as one item that determine the same health status which is not correct |
||||||
|
Line 153 |
Chinese medicine in clinics (8.8%), and Chinese medicine in hospitals (7.9%) …the difference between both is not mentioned and not clear for the reader |
||||||
|
This means that your cohort had patients diagnosed as neonatal lupus at birth ...is this correct |
||||||
|
What about between 7 to 8 ...not mentioned ?? |
||||||
|
Discussion |
The authors did not mention why they chose female patients and did not include male patients Please include studies about complementary medicine in lupus as awhole not only for specific symptoms |
||||||
Author Response
Reviewer #1, comment #1:
Abstract: Well structured and written. The conclusion is supported by the results
Response to Reviewer #1, comment #1:
We appreciate the reviewer for the time and effort in providing all the valuable comments to our manuscript.
________________________________________________________________
Reviewer #1, comment #2:
Introduction: Interesting for reading.
(1) fitness walking or strolling, (2) Buddhist prayer or attending temples, (3) vi…etc... It is better to delete numbering here as they are confusing with the reference number
Response to Reviewer #1, comment #2:
We have changed the text description into a new Table 1 to improve readability in the revised manuscript. [Page 3 to 4]
________________________________________________________________
Reviewer #1, comment #3:
Materials and Methods Study design and population: In the present study, we focused on the use of complementary therapies in female patients with different SLE clinical manifestations…this is a repition for what have mentioned in the introduction
Response to Reviewer #1, comment #3:
This statement was put in the Methods section to emphasis the differences between the present study and our previous investigation that was based on the same survey. We also added a new sentence after it to explain why only female patients were included in this study. [Line 82–84]
________________________________________________________________
Reviewer #1, comment #4:
Inclusion and exclusion criteria: The inclusion of females only was not mentioned in the authors in the inclusion criteria.
Line 92.. aged 20..why the author choose this age and not 18.
96.. major..I think this not a scientific term ..
Response to Reviewer #1, comment #4:
We have added the word “female” in the inclusion criteria. [Line 99]
Under the current Civil Code of Taiwan, the age of majority is 20 years. Therefore, we set the age to 20 years rather than 18 years.
We have changed the word “major” to “important”. [Line 103]
________________________________________________________________
Reviewer #1, comment #5:
Clinical manifestations of SLE: It will be more valuable to include the symptoms for which complementary treatment was needed.
The authors need to explain why they choose these 11 symptoms only
Including dialysis as a common clinical manifestation is not appropriate.
Response to Reviewer #1, comment #5:
The clinical symptoms included in this study were selected based on the definition of the SLE Disease Activity Index 2000 (SLEDAI-2K) and their frequency of occurrence according to Nossent et al. (Lupus 2010;19(8):949-56) and Bertsias et al. (Nat Rev Rheumatol 2013;9(11):687-94.). [Line 107–111]
We followed the suggestion by the reviewer and removed dialysis from our list of clinical manifestation of SLE and revised Figure 1 accordingly.
________________________________________________________________
Reviewer #1, comment #6:
Line 111: What is meant by age interval
Response to Reviewer #1, comment #6:
We analyzed age as a binary variable with intervals of 20 to 39 and ³ 40.
________________________________________________________________
Reviewer #1, comment #7:
Line 113: alcohol use in the past year. The authors specify the last year. Why not the alcohol in general ..and what is meant by use ..chronic drinking or only one time or what ..this point is not clear
Response to Reviewer #1, comment #7:
We used a one-year recall period to provide the respondent with a frame of reference to recall their behavior. This will avoid inconsistencies as a result of differences in the recall periods between respondents.
In the response categories of alcohol use, we have revised the term “yes” to “daily or occasionally” to clarify the meaning of drinking. [Table 2]
________________________________________________________________
Reviewer #1, comment #8:
The method of questionnaire distribution among patients was not mentioned and the method of fulfilling the questionnaire is not mentioned either by the patient or by the researcher.
Please determine if this questionnaire was validated or not.
Determine the language of questionnaire and if it was translated or not.
Response to Reviewer #1, comment #8:
A paper-based questionnaire was used to obtain information from the patients. Two research nurses of the rheumatology clinic were available to assist the completion of the questionnaire, if necessary. We have added this information in the Methods section of the revised manuscript. [Line 87–89]
The questionnaire was in Chinese and it was not translated from another language.
Since the questionnaire consists of single item questions rather than measurement scale, it is not possible to conduct validity tests, such as, exploratory factor analysis or reliability test, such as, internal consistency. Nevertheless, the content validity of the questionnaire was evaluated by experts and deemed to be valid.
________________________________________________________________
Reviewer #1, comment #9:
Line 145: average, poor, or very poor…. the authors considered average and poor as one item that determine the same health status which is not correct
Response to Reviewer #1, comment #9:
We did evaluate self-report health status with or without combining the “average” category to “poor” and “very poor” categories. The results of the multiple logistic regression analyses were the same. Self-report health status was not significantly associated with the SLE clinical manifestations.
________________________________________________________________
Reviewer #1, comment #10:
Line 153: Chinese medicine in clinics (8.8%), and Chinese medicine in hospitals (7.9%) …the difference between both is not mentioned and not clear for the reader
Response to Reviewer #1, comment #10:
To improve clarity of the terms, we have changed them to “Visit Chinese medicine clinics” and “Visit Chinese medicine department in a hospital” in the revised manuscript.
________________________________________________________________
Reviewer #1, comment #11:
Age at SLE diagnosis, years 0–29
This means that your cohort had patients diagnosed as neonatal lupus at birth ...is this correct
Response to Reviewer #1, comment #11:
We appreciate the reviewer this suggestion. We only included patients with SLE diagnosed after the age 20 years in the study. Therefore, we do not have patients with neonatal lupus. We have revised the Table 2 and 3 to correctly indicate that the age at SLE diagnosis was 20 to 29 or ³ 30 years to avoid confusion.
________________________________________________________________
Reviewer #1, comment #12:
Sleep duration/day, hours
³ 8 59 (18.6)
£ 7
What about between 7 to 8 ...not mentioned ??
Response to Reviewer #1, comment #12:
We appreciate the reviewer for raising this question. The original question on the questionnaire has four response categories (£ 5 hours, 6-7 hours, 8-9 hours, and ³ 10 hours). We collapsed the four categories into two. Respondents who sleep more than 7 hours but not up to 8 hours would simply choose the 6-7 hours category.
________________________________________________________________
Reviewer #1, comment #13:
Discussion: The authors did not mention why they chose female patients and did not include male patients.
Please include studies about complementary medicine in lupus as a whole not only for specific symptoms.
Response to Reviewer #1, comment #13:
We have added a sentence in the Methods section of the revised manuscript. Because the clinical manifestations of SLE are different between the two sexes and that there are insufficient number of male patients, and therefore, this study focused only on female patients. [Line 83–84]
Our previous study (reference 9) investigated the prevalence of and the factors associated with the regular use of complementary therapies for Taiwanese patients with systemic lupus erythematosus. We focused the discussion on the specific symptoms we found in the present study.
Reviewer 2 Report
This study examined the type of complementary therapy that the Taiwanese female SLE patients were using. I am not really sure whether the title fully reflects the contents of this paper. Because this is a study that assessed the frequency and type of complementary therapy among SLE patients, I would rather recommend the authors to revise it.
Abstract section:
- It was revealed that a certain clinical manifestation was associated with the use of specific complementary therapy. Unfortunately, I believe the conclusions of this paper cannot be supported by the results of the authors. For example, those with renal involvement had higher probability of using probiotics; could this be interpreted that probiotics are benefical in lupus nephritis? In fact, this paper only investigated the pattern of complementary therapy in patients with SLE. It will be misleading to the readership when it is stated that health care providers should have up-to-date knowledge of complementary therapy (it is not even recommended as a standard therapy for SLE) and provide evidence-based recommendations. SLE patients could have sought complementary therapy because they had severe or a treatment refractory lupus. In this context, the last sentence of this paper seems to be an overstatement.
Materials and methods section:
2.3. Clinical manifestations of SLE subsection
- The selection of these variables appears to be rather arbitrary and is not clear. For example, Raynaud phenomenon is a relatively common but a feature that is not associated with the severity of disease. On the other hand, Sicca symptom is also a common but the diagnosis of secondary Sjogren could be different from having a dry symptom. In particular, those that were included in the SLEDAI-2K (i.e. leukopenia, thrombocytopenia, low complement, elevated dsDNA level, and fever) were omitted in this investigation.
2.4. Measurement of demographic variables subsection
- Self-perceived health status: How was this investigated? Is this a validated questionnaire composed as very good and good / average, poor and very poor?
- Smoking habit, alcohol, regular exercise past-year: A more detailed description if required. For example, is intermittent smoking considered as a Yes? How did the authors classify regular exercise?
- Disease activity/severity and disease duration (as well as laboratory data of complements, dsDNA titer, leukocyte and lymphocyte count) are important factors associated with disease activity and/or health related quality-of-life. Relevant data should be provided and included throughout the analyses.
2.5. Use of complementary therapy subsection
- I would recommend the authors to provide a brief table summarizing the variables, which will be useful for the readers.
Figure 1. Do the numbers indicate percentage? It should be clearly provided. In addition, this is obviously a skewed population having a high proportion of Raynaud phenomenon and photosensitivity, while having a low proportion of renal involvement.
Table 1. Please be specific in the variables. How was underweight/overweight or obese defined?
Table 2. It is unclear what variables are included in the unadjusted analyses. Provide full variables that were used in the Table. In addition, variables used in Table 1 and 2 is not identical. Age seems to be the age when the survey was undergone (Table 1), while it is written age at diagnosis in Table 2. The variables should be consistent throughout the paper.
Line 162: It is Table 2, not Table 3.
Discussion section:
As stated above, the findings of this study should be merely interpreted as a fact. It is with insufficient evidence to state that certain complementary therapy will be beneficial in managing the clinical manifestations of SLE.
Additional comments:
English should be improved by a person who is a native English speaker.
Author Response
Reviewer #2, comment #1:
Abstract section:
- It was revealed that a certain clinical manifestation was associated with the use of specific complementary therapy. Unfortunately, I believe the conclusions of this paper cannot be supported by the results of the authors. For example, those with renal involvement had higher probability of using probiotics; could this be interpreted that probiotics are benefical in lupus nephritis? In fact, this paper only investigated the pattern of complementary therapy in patients with SLE. It will be misleading to the readership when it is stated that health care providers should have up-to-date knowledge of complementary therapy (it is not even recommended as a standard therapy for SLE) and provide evidence-based recommendations. SLE patients could have sought complementary therapy because they had severe or a treatment refractory lupus. In this context, the last sentence of this paper seems to be an overstatement.
Response to Reviewer #2, comment #1:
We appreciate the reviewer’s comment. We do not advocate the use of complementary medicine for patients with SLE. Nevertheless, based on our clinical experience, patients are using various types of complementary medicine. We believe that health care providers should be able to have a clear understanding of the safety of these treatments, and be able to provide evidence-based recommendations to patients with SLE.
We have deleted the last sentence following the reviewer’s suggestion.
________________________________________________________________
Reviewer #2, comment #2:
2.3. Clinical manifestations of SLE subsection
- The selection of these variables appears to be rather arbitrary and is not clear. For example, Raynaud phenomenon is a relatively common but a feature that is not associated with the severity of disease. On the other hand, Sicca symptom is also a common but the diagnosis of secondary Sjogren could be different from having a dry symptom. In particular, those that were included in the SLEDAI-2K (i.e. leukopenia, thrombocytopenia, low complement, elevated dsDNA level, and fever) were omitted in this investigation.
Response to Reviewer #2, comment #2:
We selected the clinical symptoms according to the studies by Nossent et al. (Lupus . 2010 Jul;19(8):949-56, Nat Rev Rheumatol. 2013;9(11):687-94.) and Bertsias et al. (Nat Rev Rheumatol. 2013;9(11):687-94.), and the definition of SLE Disease Activity Index 2000 (SLEDAI-2K). In addition, we chose those symptoms that would have a direct impact on patients’ quality of life and excluded those that do not have a direct impact on patients’ quality of life.
Because patients with SLE are more likely to seek complementary therapies for Raynaud phenomenon or Sicca symptom than for low C3 or leukopenia, the latter manifestations were not included in our study. We have revised the manuscript accordingly [line 108–117]. The association of specific type of complementary therapies with SLEDAI-2k can be found in our previous paper (BMC Complement Med Ther. 2021;21(1):247).
________________________________________________________________
Reviewer #2, comment #3:
2.4. Measurement of demographic variables subsection
- Self-perceived health status: How was this investigated? Is this a validated questionnaire composed as very good and good / average, poor and very poor?
- Smoking habit, alcohol, regular exercise past-year: A more detailed description if required. For example, is intermittent smoking considered as a Yes? How did the authors classify regular exercise?
- Disease activity/severity and disease duration (as well as laboratory data of complements, dsDNA titer, leukocyte and lymphocyte count) are important factors associated with disease activity and/or health related quality-of-life. Relevant data should be provided and included throughout the analyses.
Response to Reviewer #2, comment #3:
Self-perceived health status was asked for a Likert-type scale question with five response items (very good, good, fair, poor, very poor). The question is commonly used to assess self-perceived health status in survey research.
Smoking, alcohol use, and regular exercise were all single question items with multiple response items. We collapsed the response categories “daily” and “occasionally” to a single group. We have revised the name of the category from “yes” to “daily or occasionally” in Table 2 to clarify the meaning of the category. In the question on exercise, the respondents were asked whether they had engaged in exercise that lasted at least 20 minutes that made them breathe faster and sweat in the past year. The response categories were never, once a month or fewer, several times a month, several times a week, and daily. We collapsed the last two categories (several times a week and daily) and defined it as engaging in regular vigorous exercise. We have added this description in the Methods section. [Line 125–130].
Regarding the disease activity index, we did not include it in the regression model as an independent variable because it consists of the clinical manifestations that we evaluate as dependent variables in this study. Including the disease activity index in the model will simply spuriously inflate the R square of the model.
________________________________________________________________
Reviewer #2, comment #4:
2.5. Use of complementary therapy subsection
- I would recommend the authors to provide a brief table summarizing the variables, which will be useful for the readers
Response to Reviewer #2, comment #4:
We have changed the original descriptive text to a new Table 1 to improve the readability.
________________________________________________________________
Reviewer #2, comment #5:
Figure 1. Do the numbers indicate percentage? It should be clearly provided. In addition, this is obviously a skewed population having a high proportion of Raynaud phenomenon and photosensitivity, while having a low proportion of renal involvement.
Response to Reviewer #2, comment #5:
We have added “Numbers shown are percentages” in the footnote of Figure 1.
The distribution appeared to be skewed because our patients with SLE were recruited from the outpatient department. Only patients with active problem (e.g., daily total urine protein > 0.5 g) within the past 14 days were used in the calculation. Therefore, the proportion of renal involvement is low. Regarding the high proportion of Raynaud phenomenon and photosensitivity, we speculated that it might related to the older age (majority > 40 years) of our study sample.
________________________________________________________________
Reviewer #2, comment #6:
Table 1. Please be specific in the variables. How was underweight/overweight or obese defined?
Response to Reviewer #2, comment #6:
In this study, overweight (24 £ body mass index < 27 kg/m2) and obesity (body mass index ≥ 27 kg/m2) were defined according to the Ministry of Health and Welfare, Taiwan. We have added the above sentence in Section 2.4. [Line 123–125].
________________________________________________________________
Reviewer #2, comment #7:
Table 2. It is unclear what variables are included in the unadjusted analyses. Provide full variables that were used in the Table. In addition, variables used in Table 1 and 2 is not identical. Age seems to be the age when the survey was undergone (Table 1), while it is written age at diagnosis in Table 2. The variables should be consistent throughout the paper.
Response to Reviewer #2, comment #7:
As indicated in the data analysis section, the independent variables were the 10 types of complementary therapies with the highest prevalence (see Figure 1) and all demographic variables listed in Table 1. The list of variables is also included in the last paragraph of the Results section.
The variable “age interval” in Table 1 was entered into the multiple logistic regression model but did not retain in the model with a backward selection method based on the likelihood ratio test. However, age at SLE diagnosis was retained in the final model. This explains why only one of the two age variables remain in Table 2.
________________________________________________________________
Reviewer #2, comment #8:
Line 162: It is Table 2, not Table 3.
Response to Reviewer #2, comment #8:
Thank you for pointing the error, and we have corrected it.
________________________________________________________________
Reviewer #2, comment #9:
Discussion section:
As stated above, the findings of this study should be merely interpreted as a fact. It is with insufficient evidence to state that certain complementary therapy will be beneficial in managing the clinical manifestations of SLE.
Response to Reviewer #2, comment #9:
We agree the reviewer that the present study was not set up to provide support to the use of certain complementary therapy in the management of clinical manifestations of SLE. The objective of this study was only to evaluate the association of prevalent SLE clinical manifestations and complementary therapy use in Taiwanese female patients with SLE. In the discussion, we tried to explore why the various associations were observed among our patients, which is probably due to common beliefs that are often have weak or no clinical supporting evidence.
________________________________________________________________
Reviewer #2, comment #10:
English should be improved by a person who is a native English speaker.
Response to Reviewer #2, comment #10:
We have revised the manuscript to ensure that the text is free from typographical and grammatical errors.
Reviewer 3 Report
Dear Authors,
You have studied the use of complementary treatments in outpatients with SLE from southern Taiwan and published these data in 2021 (your source [9]). You found that normal weight patients over 40 years of age with lower disease activity were more likely to use complementary interventions.
In the current study, they chose only the female population of the same cohort without providing a rationale.
This initially leads to a selection bias.
The paper does not mention disease duration or disease activity, important clinical data. In the previous publication, you had seen no difference in disease activity (SLEDAI 2k) with the use of complementary treatments. Since renal involvement causes a major impact on activity, these data should be evaluated.
Disease harm (SLICC) is not evaluated, but the question of whether complementary medicine prevents it would be important.
On the question of prevention, it would be important to know whether interventions were used before the disease manifested or only when symptoms or damage occurred.
Thus, it is understandable that in the case of damage such as renal insufficiency as an expression of the failure of conventional therapy, an alternative is sought.
I think it would be useful to separate physical activity from the other methods. For this, other activities such as swimming or cycling could be combined with running.
The study cited in [12] deals with nonpharmacologic treatments for SLE, but it is specifically predominantly psychological interventions, which by their nature improve anxiety, depression, and fatigue. Placing them in the context of the interventions you evaluated without citing the intervention gives the false impression that other complementary interventions might affect symptoms.
In the discussion, you correctly mention the limitations (selected cohort).
Yours sincerely
Author Response
Reviewer #3, comment #1:
You have studied the use of complementary treatments in outpatients with SLE from southern Taiwan and published these data in 2021 (your source [9]). You found that normal weight patients over 40 years of age with lower disease activity were more likely to use complementary interventions.
In the current study, they chose only the female population of the same cohort without providing a rationale.
This initially leads to a selection bias.
Response to Reviewer #3, comment #1:
Only female patients were analyzed in this study because the clinical manifestations of SLE are different between the sexes. Therefore, we focused only in 317 female patients because there were only 34 male patients, which is not enough to reliably conduct a separate analyses. As we have indicated in the title of the study, readers should be clear that our findings are applicable only to female patients with SLE. Therefore, it should not lead to a selection bias. We have revised the manuscript accordingly (Line 83–84) and added a new reference.
________________________________________________________________
Reviewer #3, comment #2:
The paper does not mention disease duration or disease activity, important clinical data. In the previous publication, you had seen no difference in disease activity (SLEDAI 2k) with the use of complementary treatments. Since renal involvement causes a major impact on activity, these data should be evaluated.
Disease harm (SLICC) is not evaluated, but the question of whether complementary medicine prevents it would be important.
Response to Reviewer #3, comment #2:
Regarding the disease activity index, we did not include it in the regression model as an independent variable because it consists of the clinical manifestations that we evaluated as dependent variables in this study. Including the disease activity index in the model will simply spuriously inflate the R square of the model.
We did not collect SLE damage index. A prospective study is needed to evaluate whether a particular complementary therapy can prevent damage or harm related to SLE.
________________________________________________________________
Reviewer #3, comment #3:
On the question of prevention, it would be important to know whether interventions were used before the disease manifested or only when symptoms or damage occurred.
Thus, it is understandable that in the case of damage such as renal insufficiency as an expression of the failure of conventional therapy, an alternative is sought.
Response to Reviewer #3, comment #3:
We agree with the reviewer. However, it is not possible to delineate the temporal relationship with our cross-sectional study, which is a limitation of our study design.
________________________________________________________________
Reviewer #3, comment #4:
I think it would be useful to separate physical activity from the other methods. For this, other activities such as swimming or cycling could be combined with running.
Response to Reviewer #3, comment #4:
In the multiple logistic regression analysis, fitness walking or strolling was independently and significantly associated with Raynaud’s phenomenon. As its independent effect was assessed by the use of a multiple regression model, it is not necessary to analyze it separately.
Swimming, cycling, and running are not popular type of exercise adopted by our patients.
________________________________________________________________
Reviewer #3, comment #5:
The study cited in [12] deals with nonpharmacologic treatments for SLE, but it is specifically predominantly psychological interventions, which by their nature improve anxiety, depression, and fatigue. Placing them in the context of the interventions you evaluated without citing the intervention gives the false impression that other complementary interventions might affect symptoms.
Response to Reviewer #3, comment #5:
We appreciate the reviewer for pointing out this. We have added “predominantly psychological interventions” in our description to specify the type of interventions to avoid confusion. [Line 70–71].
________________________________________________________________
Reviewer #3, comment #6:
In the discussion, you correctly mention the limitations (selected cohort).
Response to Reviewer #3, comment #6:
We highly appreciate the time and effort that the reviewer spent in assessing our manuscript.
Round 2
Reviewer 1 Report
All reviewer comments have been covered by the authors and their work is appreciated
Author Response
Reviewer's comment: All reviewer comments have been covered by the authors and their work is appreciated.
Authors' response to reviewer's comment: We greatly appreciate the reviewer for taking the time to review our manuscript.
Reviewer 2 Report
Thank you for the response.
Author Response
Reviewer's comment: Thank you for the response.
Authors' response to reviewer's comment: We greatly appreciate the reviewer for taking the time and effort to review our manuscript.
Reviewer 3 Report
Dear authors,
thank you for the responses and modifications.
The study has some methodical limitations and generalisation should be avoided - as you do.
In your study you could not distinguish between complementary methods taken as prevention and those taken as reaction to a manifestation. E.g. probiotics might have been taken before manifestation of skin photosensitivity thus preventing photosensitivity.
The sentence Page 7 213ff In addition, a number of food items, such as garlic, ginkgo biloba, L-carnitine, inositol nicotinate, and evening primrose oil had been evaluate to increase skin blood flow or hand skin temperature, but evidence from rigorous studies is still lacking to support any food ingredients in alleviating Raynaud's phenomenon [24] should be shortened or divided int 2. In addition had been evaluate to increase is wrong..
Author Response
Reviewer's comment #3:
The sentence Page 7 213ff In addition, a number of food items, such as garlic, ginkgo biloba, L-carnitine, inositol nicotinate, and evening primrose oil had been evaluate to increase skin blood flow or hand skin temperature, but evidence from rigorous studies is still lacking to support any food ingredients in alleviating Raynaud's phenomenon [24] should be shortened or divided int 2. In addition had been evaluate to increase is wrong.
Authors' response to reviewer's comment #3:
We appreciate the reviewer’s comment and we have revised the sentence to:
“In addition, a number of food items, such as garlic, ginkgo biloba, L-carnitine, inositol nicotinate, and evening primrose oil had been reported to increase skin blood flow or hand skin temperature. Overall, evidence from rigorous studies is still lacking to support any food ingredients in alleviating Raynaud's phenomenon [22].” (page 7, line 209-213)